# Multisite surveillance for influenza and other respiratory viruses in India: 2016–2018

**Mandeep Chadha[1¤a], Aslesh Ottapura Prabhakaran[2], Manohar Lal Choudhary[1], Dipankar Biswas[3], Parvaiz Koul[4], K. Kaveri[5], Lalit Dar[6], Chawla Sarkar Mamta[7], Santosh Jadhav[1], Sumit Dutt Bhardwaj[1], Kayla Laserson[2¤b], Siddhartha Saha[2], Varsha Potdar[1]***

**1** Indian Council of Medical Research-National Institute of Virology, Pune, India, **2** US Centers for Disease Control and Prevention (India Office), New Delhi, India, **3** Indian Council of Medical Research-Regional Medical Research Centre, Dibrugarh, India, **4** Sher-i-Kashmir Institute of Medical Sciences, Srinagar, India, **5** King Institute of Preventive Medicine and Research, Chennai, India, **6** All India Institute of Medical Sciences, New Delhi, India, **7** Indian Council of Medical Research-National Institute of Cholera and Enteric Diseases, Kolkata, India

¤a Current address: Technical consultant, WHO, Geneva, Switzerland
¤b Current address: Bill and Melinda Gates Foundation, New Delhi, India
* potdarvarsha9@gmail.com

**Data Availability Statement:** The original data contributions presented in the study are included in the article/supplementary material, further inquiries/requests can be directed to the

## Abstract

There is limited surveillance and laboratory capacity for non-influenza respiratory viruses in India. We leveraged the influenza sentinel surveillance of India to detect other respiratory viruses among patients with acute respiratory infection. Six centers representing different geographic areas of India weekly enrolled a convenience sample of 5–10 patients with acute respiratory infection (ARI) and severe acute respiratory infection (SARI) between September 2016-December 2018. Staff collected nasal and throat specimens in viral transport medium and tested for influenza virus, respiratory syncytial virus (RSV), parainfluenza virus (PIV), human meta-pneumovirus (HMPV), adenovirus (AdV) and human rhinovirus (HRV) by reverse transcription polymerase chain reaction (RT-PCR). Phylogenetic analysis of influenza and RSV was done. We enrolled 16,338 including 8,947 ARI and 7,391 SARI cases during the study period. Median age was 14.6 years (IQR:4–32) in ARI cases and 13 years (IQR:1.3–55) in SARI cases. We detected respiratory viruses in 33.3% (2,981) of ARI and 33.4% (2,468) of SARI cases. Multiple viruses were co-detected in 2.8% (458/16,338) specimens. Among ARI cases influenza (15.4%) were the most frequently detected viruses followed by HRV (6.2%), RSV (5%), HMPV (3.4%), PIV (3.3%) and AdV (3.1%),. Similarly among SARI cases, influenza (12.7%) were most frequently detected followed by RSV (8.2%), HRV (6.1%), PIV (4%), HMPV (2.6%) and AdV (2.1%). Our study demonstrated the feasibility of expanding influenza surveillance systems for surveillance of other respiratory viruses in India. Influenza was the most detected virus among ARI and SARI cases.

corresponding author and the deidentified data can be shared.

**Funding:** This study was funded by Department of Health and Human Services-Centers for Disease Control and Prevention (DHHS-CDC) Cooperative agreement number NU2GGH001903 and Indian Council of Medical Research-National Institute of Virology, Pune. Authors did not receive salary from this funding. DHHS-CDC was involved in study design, analysis and drafting of manuscript.

**Competing interests:** The authors have declared that no competing interests exist.

## Introduction

Respiratory viruses like influenza and respiratory syncytial virus (RSV) are responsible for substantial global morbidity and mortality annually, with substantial burden shared by young children and older adults [1–3]. While many countries have sentinel surveillance generating data on epidemiology of influenza, data about other respiratory viruses (ORV) are scarce especially in lower-Middle income countries (LMIC) although they are estimated to cause substantial morbidity and mortality [4, 5]. Infection with ORV usually presents with influenza-like-illness with symptoms like cough with or without fever but exhibits differences in epidemiology, severity and seasonality [5]. In view of India's geographic location, vast population and diverse seasonality, it is crucial that the influenza surveillance system be strengthened to detect ORVs. Understanding epidemiology of influenza and ORV in India would help in timing the use of empirical antiviral treatment for influenza, influenza vaccination, and implementing non-pharmacological interventions.

The WHO Global Influenza Surveillance and Response System (GISRS) has been monitoring influenza viruses for more than six decades. GISRS operates through a global network of National Influenza Centers (NIC), which are laboratories that provide information on the circulation and evolution of influenza viruses globally. In 2004, US Centers for Disease Control and Prevention (CDC) started collaborating with Indian Council of Medical Research—National Institute of Virology, to strengthen a network of 10 surveillance centers across India [6]. Annual percent positivity of influenza in influenza like illness(ILI)/ severe acute respiratory infection (SARI) specimens was found to be between 12–14%, but the prevalence of ORV was unknown [7]. The emergence of the COVID-19 pandemic has further highlighted the need for surveillance for ORV. WHO has been evaluating the feasibility of leveraging the GISRS platform for surveillance of COVID-19, RSV and other common respiratory viruses [8].

In order to strengthen the capacity for pandemic influenza preparedness and response, as well as to assess the feasibility of conducting surveillance for multiple respiratory viruses and understand their epidemiology, the Indian Council of Medical Research-National Institute of Virology (ICMR-NIV), in collaboration with US Centers for Disease Control and Prevention (CDC) implemented a multi-site sentinel surveillance system for severe acute respiratory infection (SARI) and acute respiratory infection (ARI) surveillance in India from 2016 to 2018. We present the results of this multi-site pan-respiratory viral surveillance network.

## Methods

### Study setting

The acute respiratory infection surveillance network included six centers specifically selected to represent different climates and geographic areas of India. From north to south, the participating centers were Sher-i-Kashmir Institute of Medical Sciences (SKIMS), Srinagar (34.0˚N, Jammu and Kashmir); All India Institute of Medical Sciences (AIIMS) (28.6˚N, New Delhi); Indian Council of Medical Research (ICMR)-Regional Medical Research Center (RMRC), Dibrugarh (27.5˚N, Assam); ICMR-National Institute for Cholera and Enteric Diseases (NICED), Kolkata (22.6˚N, West Bengal); ICMR-National Institute of Virology (ICMR-NIV), Pune (18.5˚N, Maharashtra); King Institute of Preventive Medicine & Research (KIPMR), Chennai (13.1˚N, Tamil Nadu). ICMR-NIV, Pune was the reference and coordinating center for the study. Each study center selected 2 to 3 sentinel hospitals and clinics having general medicine and pediatrics departments for enrollment of the participants.

## Ethics approval

The study protocol was approved by the appropriate institutional human ethics committees and approved by the Health Ministry's Screening Committee of India. The participants were informed about the study in their local language and written consent/assent was obtained before enrollment in the study. For participant under 18 years of age, written informed consent was obtained from the parent/guardian.

## Duration of surveillance

Surveillance at Srinagar, Dibrugarh, Pune, and Chennai was conducted from September 2016 to December 2018. Kolkata conducted surveillance from October 2016 to October 2017, and Delhi conducted surveillance from September 2016 to September 2018.

## Case definitions

For enrolling the participants, acute respiratory infection (ARI) was defined as illness in a person presenting in the outpatient department (OPD) with acute onset (within 7 days) of any two of the following symptoms: fever/feverishness/chills, cough, nasal congestion, shortness of breath, or sore throat. Severe acute respiratory infection (SARI) definition was adapted from the WHO case definition and was defined as cases with history of cough with onset within the last 7 days and requiring overnight hospitalization. For defining SARI among infants aged <2 months, physician diagnosis suggestive of acute lower respiratory infection (pneumonia, bronchitis, bronchiolitis, sepsis) requiring overnight hospitalization was used.

## Participant enrollment and case identification

Each center enrolled a convenience sample of 5 to 10 ARI cases and 5 to 10 SARI cases of all ages every week. During any up surge in respiratory infections noticed by the clinicians in the sentinel hospitals, additional participants were enrolled. At all sentinel sites, physicians and nurses were trained to screen patients using ARI and SARI case definitions. ARI cases were screened from outpatient facility and SARI cases from general medicine, pediatric and pulmonary medicine wards. Trained staff visited outpatients clinics and inpatient wards on fixed day of week. Patients fulfilling the case definitions and consenting to participate were recruited in the study. Every week first 5–10 ARI/ SARI cases identified in outpatient clinics and inpatient wards of each center were enrolled into the study. The clinical and epidemiological details of each enrolled participant were recorded in a standardized case report form.

## Specimen collection

Clinical sample collection, transportation, and storage were done as per WHO guidelines [9]. Nasal and/or throat respiratory specimens were collected from enrolled ARI and SARI cases by trained personnel. Only nasal specimens were collected from infants aged less than 1year. Specimens were transported in viral transport media to the respective site laboratory within 24 hours in cold box with ice packs. If the samples were not tested immediately, they were stored at -80°C.

## Virus detection

All centers performed external molecular quality assurance for the detection of respiratory viral pathogens (provided by the ICMR-NIV Pune) successfully. NIV participated in the external quality assurance program by Quality Control for Molecular Diagnostics (QCMD) for influenza and non-influenza respiratory viruses with 100% concordance. All the centers

followed standard operating procedures for sample processing and viral detection using the same RTPCR assays. RNA was extracted using a Qiagen viral RNA isolation kit by all the centers except NIV, which used a MagMax-96 kit as per manufacturer's protocol. All the specimens were tested by real-time reverse transcription polymerase chain reaction (rRT-PCR) for the following viruses: influenza A [A(H1N1)pdm09, A(H3N2)], influenza B [B/Yamagata and B/Victoria lineages] along with house-keeping RNaseP gene, respiratory syncytial virus A and B, metapneumovirus, parainfluenza viruses 1, 2, 3, and 4, rhinoviruses and adenoviruses using a protocol previously described [10]. Nucleic acid amplification was performed using one step RT-PCR (qRT-PCR SuperScript III kit, Invitrogen, USA). A 25 μl PCR reaction comprised of 10 μmol of each forward and reverse primer, 5 μmol of TaqMan probe, 12.5 μl 2X buffer, 0.5 μl SuperScript III enzyme and 5 μl nucleic acid templates. Thermal cycling conditions were: 50˚C for 30 minutes for reverse transcription, initial denaturation at 94˚C for 5 minutes, 45 cycles of three steps (15 seconds at 94˚C, 15 seconds at 50˚C and 30 seconds at 55˚C incubation step during which fluorescence data were collected).

For identifying an oseltamivir-resistant influenza A(H1N1)pdm09 virus possessing the H275Y mutation (in the neuraminidase (NA) gene) in clinical specimens or in clinical isolates, the laboratory performed allelic discrimination using a rRT-PCR protocol shared by the National Institute of Health, Thailand [11].

Sequencing of the HA gene of influenza and G gene of RSV was carried on a subset of positive samples using ABI 3730 DNA analyzer [12]. The sequence obtained was edited by Seqscape V2.5 software (Applied Biosystems, USA), and pairwise sequence alignment and phylogeny of HA of influenza viruses and G gene of RSV were performed using best fit Tamura-Nei nucleotide substitution model to generate a neighbor-joining tree with 1000 replicates bootstrap support, representing reference strains, and previously reported strains from India and strains from the global database using MEGA 6 program [13].

### Data analysis

All data were entered into Epi-info 7 by each participating center and collated every week at NIV, Pune and analyzed using STATA 15 (StataCorp LLC). The overall percent positivity for each virus for surveillance period was calculated as proportion of ARI and SARI samples tested positive for each virus along with 95% confidence interval. Monthly percent positivity (mPP) of each virus for each center were calculated as a proportion of specimens tested positive out of all specimens (ARI and SARI) with symptom onset in same month. To detect increase in activity we compared mPP with overall percent positivity during the project period (September 2016 to October 2018). Months with mPP higher than percent positivity were considered as having increased virus activity. Chi-square test was used to measure statistical significance of virus positivity across different groups. Multivariable logistic regression was used to derive adjusted odds ratios after adjusting for age and center.

### Results

We collected respiratory specimens from 8,947 ARI cases (Table 1) and 7,391 SARI cases (Table 2) from six centers between September 2016 to December 2018. Median age of subjects sampled was 14.6 years (IQR:4–32) in ARI cases and 13 years (IQR:1.3–55) in SARI cases. Among the enrolled, 2,649 (29.6%) ARI cases and 2,955 (40.0%) SARI cases were aged < 5 years. Proportion of cases under 5 years was lowest in Pune (11.7%) among ARI cases and in Srinagar (7.6%) among SARI cases. Proportion cases > = 60 was highest in Delhi (21.9%) among ARI cases and in Srinagar (53.2%) among SARI cases. In ARI cases, 4,220 (47.2%) were female and in SARI 3,191 (43.2%) were female.

**Table 1. Percent positive of respiratory viruses by clinical-epidemiological characteristic among outpatients with acute respiratory infection, India 2016–2018.**

| | No. tested | Any respiratory virus | | Influenza | | RSV | |
|---|---|---|---|---|---|---|---|
| | | N (%) | Adj OR# (95% CI) | N (%) | Adj OR # (95% CI) | N (%) | Adj OR # (95% CI) |
| **All tested** | 8947 | 2981 (33.3) | | 1374 (15.4) | | 445 (5.0) | |
| **Location** | | | | | | | |
| Chennai* | 1,491 | 259 (17.4) | - | 146 (9.8) | - | 39 (2.6) | - |
| Delhi | 1,217 | 375 (30.8) | 2.4 (2–2.9) | 207 (17) | 2.1 (1.6–2.6) | 53 (4.4) | 1.8 (1.2–2.8) |
| Dibrugarh | 2,149 | 775 (36.1) | 2.2 (1.9–2.6) | 230 (10.7) | 1.1 (0.9–1.4) | 145 (6.7) | 1.9 (1.3–2.7) |
| Kolkata | 929 | 419 (45.1) | 1.1 (0.9–1.4) | 105 (11.3) | 1 (0.8–1.2) | 8 (0.9) | 1.3 (0.8–2) |
| Pune | 1,479 | 259 (17.5) | 3.1 (2.6–3.7) | 141 (9.5) | 1.1 (0.9–1.5) | 39 (2.6) | 0.2 (0.1–0.5) |
| Srinagar | 1,682 | 894 (53.2) | 6.7 (5.6–7.9) | 545 (32.4) | 5.0 (4.1–6.2) | 161 (9.6) | 5 (3.5–7.2) |
| **Age group** | | | | | | | |
| <6 month | 190 | 75 (39.5) | 4 (2.7–5.8) | 9 (4.7) | 0.7 (0.3–1.4) | 23 (12.1) | 10.4 (4.9–22.2) |
| 6months -<2years | 1,050 | 474 (45.1) | 5.1 (4–6.6) | 96 (9.1) | 1.5 (1.1–2.2) | 104 (9.9) | 9 (4.7–17.1) |
| 2–4 years | 1,409 | 668 (47.4) | 5.9 (4.6–7.5) | 256 (18.2) | 3.4 (2.5–4.6) | 109 (7.7) | 6.9 (3.6–13) |
| 5–14 years | 1,841 | 567 (30.8) | 3.3 (2.6–4.2) | 322 (17.5) | 3.4 (2.5–4.5) | 68 (3.7) | 3.2 (1.7–6.2) |
| 15–59 years | 3,789 | 1092 (28.8) | 2.3 (1.8–2.9) | 631 (16.7) | 2.2 (1.6–2.9) | 130 (3.4) | 2.1 (1.1–3.9) |
| > = 60 years* | 668 | 105 (15.7) | - | 60 (9) | - | 11 (1.6) | - |
| **Gender** | | | | | | | |
| Male* | 4,727 | 1570 (33.2) | - | 714 (15.1) | - | 244 (5.2) | - |
| Female | 4,220 | 1411 (33.5) | 1 (0.9–1.1) | 660 (15.6) | 1 (0.9–1.1) | 201 (4.8) | 0.9 (0.8–1.1) |
| **Period** | | | | | | | |
| Sep 2016-Aug 2017* | 4628 | 1846 (39.9) | - | 810 (17.5) | - | 249 (5.4) | - |
| Sep 2017- Dec 2018 | 4319 | 1135 (26.3) | 0.7(0.6–0.8) | 564 (13.1) | 0.8 (0.7–0.9) | 196 (4.5) | 2.0 (1.7–2.5) |
| **Pregnancy** | | | | | | | |
| Pregnant female | 172 | 74 (43.0) | 1.4 (1.0–2.1) | 45 (26.2) | 1.2 (0.8–1.7) | 16 (9.3) | 2.3 (1.2–4.2) |
| Non pregnant female 15–44 years* | 1,487 | 400 (26.9) | - | 222(14.3) | - | 45 (3.0) | - |
| **Pre-existing medical conditions** | | | | | | | |
| At least one | 1,018 | 280 (27.5) | 0.8 (0.7–0.9) | 126 (12.4) | 0.6 (0.5–0.8) | 31 (3) | 0.8 (0.5–1.1) |
| None* | 7,929 | 2701 (34.1) | - | 1248 (15.7) | - | 414 (5.2) | - |
| **Symptoms** | | | | | | | |
| **Fever** | | | | | | | |
| **Present** | 7974 | 2794 (35) | 2.1 (1.8–2.5) | 1338 (16.8) | 4.8 (3.4–6.7) | 431 (5.4) | 2.9 (1.7–5.1) |
| **Absent*** | 973 | 187 (19.2) | | 36 (3.7) | | 14 (1.4) | |
| **Cough** | | | | | | | |
| **Present** | 8374 | 2817 (33.6) | 1.1 (0.9–1.4) | 1300 (15.5) | 1.2 (0.9–1.5) | 420 (5) | 1 (0.7–1.5) |
| **Absent*** | 573 | 164 (28.6) | | 74 (12.9) | | 25 (4.4) | |
| **Sore throat** | | | | | | | |
| **Present** | 5587 | 1685 (30.2) | 1 (0.9–1.2) | 858 (15.4) | 1.1 (0.9–1.2) | 249 (4.5) | 1.1 (0.9–1.4) |
| **Absent*** | 3360 | 1296 (38.6) | | 516 (15.4) | | 196 (5.8) | |
| **Nasal discharge** | | | | | | | |
| **Present** | 6872 | 2462 (35.8) | 1.3 (1.1–1.4) | 1114 (16.2) | 1.3 (1.1–1.5) | 365 (5.3) | 1 (0.8–1.3) |
| **Absent*** | 2075 | 519 (25) | | 260 (12.5) | | 80 (3.9) | |
| **Breathlessness** | | | | | | | |
| **Present** | 2918 | 1013 (34.7) | 1 (0.9–1.1) | 509 (17.4) | 1 (0.8–1.1) | 136 (4.7) | 0.9 (0.7–1.2) |
| **Absent*** | 6029 | 1968 (32.6) | | 865 (14.3) | | 309 (5.1) | |
| **Malaise/fatigue** | | | | | | | |
| **Present*** | 2194 | 914 (41.7) | 1 (0.9–1.1) | 566 (25.8) | 1.7 (1.5–2) | 101 (4.6) | 0.9 (0.7–1.3) |
| **Absent** | 6753 | 2067 (30.6) | | 808 (12) | | 344 (5.1) | |

*(Continued)*

**Table 1.** (Continued)

| | No. tested | Any respiratory virus | | Influenza | | RSV | |
|---|---|---|---|---|---|---|---|
| | | N (%) | Adj OR# (95% CI) | N (%) | Adj OR # (95% CI) | N (%) | Adj OR # (95% CI) |
| **Diarrhea/Vomiting** | | | | | | | |
| **Present*** | 2361 | 971 (41.1) | 1.2 (1–1.3) | 487 (20.6) | 1.4 (1.3–1.6) | 131 (5.5) | 1 (0.8–1.2) |
| **Absent** | 6586 | 2064 (31.3) | | 887 (13.5) | | 314 (4.8) | |

#Adjusted for age and Center

* Reference group

Across all centers, respiratory viruses were detected in 2,981 (33.3%) ARI and 2,468 (33.4%) SARI cases. Virus detection in ARI cases ranged from 17.4% in Chennai to 53.2% in Srinagar (Table 1), while in SARI, virus detection ranged from 27.7% in Srinagar to 41.6% in Delhi (Table 2). Viral detection was highest among children aged 2–4 years (47.4% in ARI and 45.1% in SARI cases) and lowest in adults ≥60 years (15.7% in ARI and 23.5% in SARI cases).

## Influenza

Influenza was the most common virus detected among ARI (15.4%; 95% CI: 14.5–16.2) and SARI (12.7%; 95% CI:11.9–13.5) cases (Table 3). Influenza percent positivity among ARI cases ranged from 9.8% in Pune to 32.4% in Srinagar (Table 1). Among SARI cases, influenza percent positivity ranged from 6.4% in Dibrugarh to 15.4% in Delhi (Table 2). Influenza was the most prevalent virus in ARI and SARI cases of age group 5–15 years, 15–59 years, and ≥ 60 years (Fig 1). Influenza positivity in SARI was similar or lower than in ARI cases in all age groups except in adults ≥60 years (ARI- 9%, SARI-12.7%, p<0.01). Influenza detection was higher among pregnant women with ARI (26.2%, 95% CI:19.8–22.4)) when compared to non-pregnant women in age group 15–44 years (15.0%, 95% CI: 13.2–16.9) (p<0.01). In pregnant women with SARI influenza was detected in 27.1% (95% CI:16.3–40.3) of cases.

Influenza A(H1N1)pdm09 virus detection was higher in SARI cases (7.9%, 95% CI:7.3–8.5%) than in ARI cases (6.1%, 95%CI:5.6–6.5%) (Table 3). Influenza A(H3N2) virus detection was lower in SARI cases (2.5%, 95% CI:2.2–2.9%) when compared to ARI cases (3.9%, 95% CI:3.5–4.3%). Influenza B virus positivity was also low in SARI cases (2.3%, 95% CI:1.9–2.6%) when compared to ARI cases (5.4%, 95% CI:4.9–5.9%). Although both the lineages of influenza B were detected during the surveillance period, the most frequently detected B lineage was B/Victoria (91%) in 2017 and B/Yamagata (80%) in 2018.

The overall percent positivity for influenza A (H1N1)pdm09 ranged from 3.3% in Dibrugarh to 10.1% in Pune (S1 Table). Similarly, the overall percent positivity for the influenza A (H3N2) ranged from 0.9% in Pune to 7.3% in Srinagar, and for influenza B ranged from 0.1% in Kolkata to 8.2% in Srinagar. The mPP of influenza viruses exceeded the overall prevalence during different months at different centers (Fig 2).

Phylogenetic analysis of the HA gene of 2017–18 influenza A(H1N1)pdm09 viruses (2017 n = 62; Delhi 5, Kolkata 3, Chennai 19, Srinagar 3, Dibrugarh 10 and Pune 22; 2018 n = 55 Pune] showed that 2017–18 isolates grouped in clade 6B.1 with S84N, S162N and I216T signature amino acid change. The 2017 strains were similar to A/Michigan/45/2015 strain whereas 2018 strains were similar to A/Brisbane/02/2018 which were 2018–19 and 2019–2020 northern hemisphere vaccine components. Phylogenetic analysis of HA gene (n = 39) of influenza A/H3N2 virus grouped in Clade 3C.2a1 with signature amino acid change N171K, F159Y, N144S and similar to A/Singapore/INFIMH-16-0019/2016, which was the 2018–19 Northern

**Table 2. Percent positive of respiratory viruses by clinical-epidemiological characteristic among hospitalized patients with severe acute respiratory infection, India 2016–2018.**

| | No. tested | Any respiratory virus | | Influenza | | RSV | |
|---|---|---|---|---|---|---|---|
| | | N (%) | Adj OR# (95% CI) | N (%) | Adj OR# (95% CI) | N (%) | Adj OR# (95% CI) |
| **All tested** | 7,391 | 2468 (33.4) | | 938 (12.7) | | 605 (8.2) | |
| Location | | | | | | | |
| Chennai* | 910 | 272 (29.9) | - | 79 (8.7) | - | 107 (11.8) | - |
| Delhi | 1,325 | 551 (41.6) | 2.1 (1.8–2.6) | 204 (15.4) | 1.7 (1.3–2.2) | 110 (8.3) | 1 (0.8–1.4) |
| Dibrugarh | 626 | 189 (30.2) | 0.8 (0.7–1.1) | 40 (6.4) | 1 (0.6–1.4) | 52 (8.3) | 0.4 (0.3–0.6) |
| Kolkata | 162 | 45 (27.8) | 1.6 (1.3–1.9) | 17 (10.5) | 1.8 (1.4–2.3) | 1 (0.6) | 0.8 (0.6–1.1) |
| Pune | 2,334 | 847 (36.3) | 1.1 (0.8–1.6) | 342 (14.7) | 1.2 (0.7–2.1) | 198 (8.5) | 0.1 (0–0.4) |
| Srinagar | 2,034 | 564 (27.7) | 1.8 (1.5–2.2) | 256 (12.6) | 1.3 (1–1.7) | 137 (6.7) | 1.9 (1.4–2.7) |
| Age groups | | | | | | | |
| <6 month | 761 | 302 (39.7) | 2.8 (2.2–3.4) | 24 (3.2) | 0.2 (0.1–0.3) | 153 (20.1) | 12 (8.3–17.3) |
| 6months -<2years | 1,316 | 605 (46) | 3.6 (3.0–4.3) | 133 (10.1) | 0.8 (0.6–1) | 205 (15.6) | 8.4 (5.9–12) |
| 2–4 years | 878 | 396 (45.1) | 3.1 (2.5–3.8) | 132 (15) | 1.1 (0.9–1.5) | 96 (10.9) | 5.1 (3.5–7.5) |
| 5–14 years | 816 | 254 (31.1) | 1.6 (1.3–2.0) | 118 (14.5) | 1 (0.8–1.4) | 27 (3.3) | 1.4 (0.9–2.3) |
| 14–59 years | 1,999 | 530 (26.5) | 1.2 (1.0–1.5) | 325 (16.3) | 1.3 (1–1.6) | 59 (3) | 0.7 (0.5–1.2) |
| > = 60 years* | 1,621 | 381 (23.5) | - | 206 (12.7) | - | 65 (4) | - |
| Gender | | | | | | | |
| Males* | 4,200 | 1440 (34.3) | - | 516 (12.3) | - | 354 (8.4) | - |
| Females | 3,191 | 1028 (32.2) | 1 (0.9–1.1) | 422 (13.2) | 1 (0.9–1.2) | 251 (7.9) | 1.1 (0.9–1.3) |
| Period | | | | | | | |
| Sep 2016-Aug 2017* | 2,965 | 1003 (33.8) | - | 414 (14) | - | 160 (5.4) | - |
| Sep 2017- Dec 2018 | 4,426 | 1465 (33.1) | 1.0(0.9–1.1) | 524 (11.8) | 0.9 (0.7–1) | 445 (10.1) | 0.9 (0.8–1.1) |
| Pregnancy | | | | | | | |
| Pregnant female | 59 | 20 (33.9) | 1.3 (0.7–2.4) | 16 (27.1) | 1.5 (0.8–2.8) | 1 (1.7) | 1.2 (0.1–10) |
| Non pregnant female 15–44 years* | 525 | 133 (25.3) | - | 91 (17.3) | - | 9(1.7) | - |
| Pre-existing conditions | | | | | | | |
| At least one | 2,933 | 861 (29.4) | 0.9 (0.8–1) | 363 (12.4) | 0.7 (0.6–0.9) | 152 (5.2) | 0.7 (0.6–0.9) |
| None* | 4,458 | 1607 (36) | - | 575 (12.9) | - | 453 (10.2) | - |
| Symptoms | | | | | | | |
| Fever | | | | | | | |
| **Present** | 6662 | 2261 (33.9) | 1.2 (1–1.5) | 889 (13.3) | 2.1 (1.5–2.8) | 551 (8.3) | 1.2 (0.9–1.7) |
| **Absent*** | 729 | 207 (28.4) | - | 49 (6.7) | - | 54 (7.4) | - |
| Cough | | | | | | | |
| **Present** | 6522 | 2246 (34.4) | 1.8 (1.5–2.1) | 858 (13.2) | 1.8 (1.4–2.3) | 555 (8.5) | 1.6 (1.1–2.1) |
| **Absent*** | 869 | 222 (25.5) | - | 80 (9.2) | - | 50 (5.8) | - |
| Sore throat | | | | | | | |
| **Present** | 2547 | 806 (31.6) | 1.2 (1.1–1.3) | 400 (15.7) | 1.4 (1.2–1.6) | 133 (5.2) | 0.9 (0.7–1.1) |
| **Absent*** | 4844 | 1662 (34.3) | - | 538 (11.1) | - | 472 (9.7) | - |
| Nasal discharge | | | | | | | |
| **Present** | 3600 | 1353 (37.6) | 1.3 (1.2–1.4) | 475 (13.2) | 1.3 (1.1–1.5) | 338 (9.4) | 1 (0.8–1.2) |
| **Absent*** | 3791 | 1115 (29.4) | - | 538 (14.2) | - | 267 (7) | - |
| Breathlessness | | | | | | | |
| **Present** | 5198 | 1626 (31.3) | 0.9 (0.8–1) | 596 (11.5) | 0.7 (0.6–0.8) | 405 (7.8) | 0.9 (0.7–1) |
| **Absent*** | 2193 | | - | 342 (15.6) | - | 200 (9.1) | - |
| Malaise/fatigue | | | | | | | |
| **Present** | 3079 | 871 (28.3) | 1 (0.9–1.2) | 458 (14.9) | 1.3 (1.1–1.5) | 136 (4.4) | 0.7 (0.5–0.9) |

*(Continued)*

**Table 2.** (Continued)

| | No. tested | Any respiratory virus | | Influenza | | RSV | |
|---|---|---|---|---|---|---|---|
| | | N (%) | Adj OR# (95% CI) | N (%) | Adj OR# (95% CI) | N (%) | Adj OR# (95% CI) |
| Absent* | 4312 | 1597 (37) | - | 480 (11.1) | - | 469 (10.9) | - |
| **Diarrhea/Vomiting** | | | | | | | |
| **Present** | 2497 | 824 (33) | 1 (0.9–1.1) | 337 (13.5) | 1.1 (0.9–1.3) | 178 (7.1) | 0.8 (0.7–1) |
| **Absent*** | 4894 | 1644 (33.6) | - | 601 (12.3) | - | 427 (8.7) | - |
| **Wheeze** | | | | | | | |
| Present | 2925 | 1030 (35.2) | 1.1 (1–1.2) | 324 (11.1) | 0.8 (0.7–0.9) | 309 (10.6) | 1.4 (1.2–1.7) |
| Absent* | 4466 | 1438 (32.2) | - | 614 (13.7) | - | 296 (6.6) | - |
| **ICU admission** | | | | | | | |
| Yes | 2,713 | 924 (34.1) | 0.9 (0.8–1) | 329 (12.1) | 0.8 (0.7–0.9) | 264 (9.7) | 1.1 (0.9–1.4) |
| No* | 4,678 | 1544 (33) | - | 609 (13) | - | 341 (7.3) | - |
| **Mechanical ventilation** | | | | | | | |
| Yes | 814 | 284 (34.9) | 0.8 (0.7–1) | 112 (13.8) | 0.9 (0.8–1.1) | 65 (8) | 0.9 (0.7–1.1) |
| No* | 6577 | 2184 (33.2) | - | 826 (12.6) | - | 540 (8.2) | - |
| **Oxygen administration** | | | | | | | |
| Yes | 4463 | 1422 (31.9) | 0.8(0.7–0.9) | 512 (11.5) | 0.6(0.5–0.8) | 363 (8.1) | 0.9 (0.7–1.1) |
| No* | 2928 | 1046 (35.7) | - | 426 (14.5) | - | 242 (8.3) | - |

#Adjusted for age and site

* Reference group

Hemisphere vaccine component. Phylogenetic analysis of HA gene of influenza B Yamagata (n = 12) isolates were grouped in Y3 Phuket clade and influenza B Victoria (n = 19) isolates grouped in V1a clade, which were similar to B/Phuket/3073/2013 and Colorado/06/2017 vaccine components respectively.

Assessment of NA inhibitor susceptibility of 1,127 influenza A(H1N1)pdm09 viruses showed that 6 clinical isolates had H275Y mutations and had increased half maximal inhibitory concentration ($IC_{50}$) values in phenotypic assays conferring reduced susceptibility to oseltamivir. Of these, 3 were detected in Pune, and one each was detected in Chennai, Srinagar, and Delhi.

## Respiratory Syncytial Virus

Significantly higher detections of RSV were observed among SARI (8.2%, 95% CI:7.5–8.8) compared to ARI (5%, 95%CI:4.5–5.4%) cases (p<0.01). RSV was the most common virus detected in SARI cases aged <2 years (0-6month-15.6%, 6months to 2years-10.9%) (Fig 1). Among SARI cases, RSV percent positivity ranged from 0.6% in Kolkata to 11.8% in Chennai, and in ARI cases, RSV detection ranged from 0.9% in Kolkata to 9.6% in Srinagar. RSV-B predominated throughout the study period: 2016 (61%), 2017 (62%) and 2018 (89%).

Overall percent positivity of RSV ranged from 0.8% in Kolkata to 8% in Srinagar. Increased RSV activity with mPP more than percent positivity was seen between August to November in Delhi in 2016, 2017 and 2018, in Chennai in 2017 and 2018, in Pune in 2017 and 2018, Kolkata in 2017, and Dibrugarh in 2016 (S1 Table) (Fig 2). In 2018, increased RSV activity was seen in Dibrugarh from February to August.

Phylogenetic analysis of G gene of 27 RSVA detected in 2016, and 29 RSV B detected in 2017–18 showed that RSV A of ON1 and RSV B of BA9 genotype were in circulation.

**Table 3. Virus detection among cases of acute respiratory infection (ARI) and severe acute respiratory infection (SARI), India 2016–2018.**

| | ARI (N = 8947) | | SARI (N-7391) | |
|---|---|---|---|---|
| | **Number positive** | **Percent positive (95% CI)** | **Number positive** | **Percent positive (95% CI)** |
| Any Influenza[#] | 1374 | 15.4 (14.5–16.2) | 938 | 12.7 (11.9–13.5) |
| A(H1N1pdm09) | 543 | 6.1 (5.6–6.5) | 584 | 7.9 (7.3–8.5) |
| A(H3N2) | 352 | 3.9 (3.5–4.3) | 187 | 2.5 (2.2–2.9) |
| Any Influenza B | 482 | 5.4 (4.9–5.9) | 168 | 2.3 (1.9–2.6) |
| B (Victoria) | 342 | 3.8 (3.4–4.2) | 62 | 0.8 (0.6–1) |
| B (Yamagata) | 110 | 1.2 (1–1.5) | 94 | 1.3 (1–1.5) |
| Any Respiratory Syncytial Virus (RSV)[##] | 445 | 5 (4.5–5.4) | 605 | 8.2 (7.5–8.8) |
| RSV A | 161 | 1.8 (1.5–2.1) | 127 | 1.7 (1.4–2) |
| RSV B | 289 | 3.2 (2.9–3.6) | 485 | 6.6 (6–7.1) |
| Human meta-pneumo virus | 304 | 3.4 (3–3.8) | 193 | 2.6 (2.3–3) |
| Any Para influenza virus (PIV)[###] | 298 | 3.3 (3–3.7) | 298 | 4.0 (3.6–4.5) |
| PIV-1 | 70 | 0.8 (0.6–1) | 61 | 0.8 (0.6–1) |
| PIV-2 | 27 | 0.3 (0.2–0.4) | 22 | 0.3 (0.2–0.4) |
| PIV 3 | 139 | 1.6 (1.3–1.8) | 165 | 2.2 (1.9–2.6) |
| PIV-4 | 71 | 0.8 (0.6–1) | 67 | 0.9 (0.7–1.1) |
| Adenovirus | 275 | 3.1 (2.7–3.4) | 158 | 2.1 (1.8–2.5) |
| Rhinovirus | 552 | 6.2 (5.7–6.7) | 452 | 6.1 (5.6–6.7) |
| Multiple virus detection | 269 | 3 (2.7–3.4) | 190 | 2.6 (2.2–2.9) |
| Any virus infection | 2981 | 33.3 (32.1–34.5) | 2468 | 33.4 (32.1–34.7) |

\# Co-detection of influenza type/subtype found in 2 ARI and 3 SARI cases.

\#\# Co-detection of RSV A/RSV B found in 5 ARI and 7 SARI cases.

\#\#\# Co-detection of PIV 1/2/3/4 found in 9 ARI and 17 SARI cases

## Other Respiratory Viruses

PIV(1–4) were detected in 3.3% (95%CI:3.0–3.7%) of ARI and 4% (95%CI:3.6–4.5%) SARI cases (Table 3). PIV detection ranged from 0.7% in Chennai to 4.9% in Srinagar among ARI, and 2.6% in Srinagar to 6.8% in Delhi among SARI cases. PIV detection were higher among children aged < 5 years when compared to other age groups (Fig 1) in both ARI (<5 year-6.4% vs ≥5 year-2.1%, p value<0.01) and SARI cases (<5 year-6.5% vs ≥5 year -2.4%, p value<0.01). Among PIV subtypes, PIV-3 contributed 46.6% (95% CI: 40.9–52.5) of PIV detection in ARI and 55.3% (95% CI:49.5–61.1) of PIV detection in SARI. In 2017, PIV mPP was higher than overall percent positivity from January to May in Chennai, Pune, Kolkata, Dibrugarh, Delhi and Srinagar and also from August to September in Delhi (S1 Table).

HMPV was detected in 3.4% (95% CI; 3.0–3.7%) of ARI and 2.6% of SARI cases (95% CI:2.3–3.0%). HMPV detection ranged from 1.3% in Delhi to 13.1% in Kolkata among ARI cases and 1.9% in Srinagar to 5.6% in Kolkata among SARI. HMPV detection was higher among those aged < 5 years when compared to older age groups in both ARI (<5 year-6.4% vs ≥5 year-2.1%, p value<0.01) and SARI cases (<5 year-4.4% vs ≥5 year -1.4%, p value<0.01).

Rhinovirus (HRV) was detected in 6.2% (95%CI: 5.7–6.7%) of ARI and 6.1% (95%CI: 5.6–6.7%) in SARI cases. HRV detection ranged from 1.5% in Pune to 11.5% in Kolkata among ARI cases and 3% in Chennai to 10% in Delhi among SARI. HRV were the most common viruses detected in children aged 6months to 2 years with ARI (Fig 1). Adenoviruses (AdV) were detected in 3.1% (95% CI:2.7–3.4%)of ARI cases and 2.1% (95% CI:1.8–2.5%)of SARI cases.

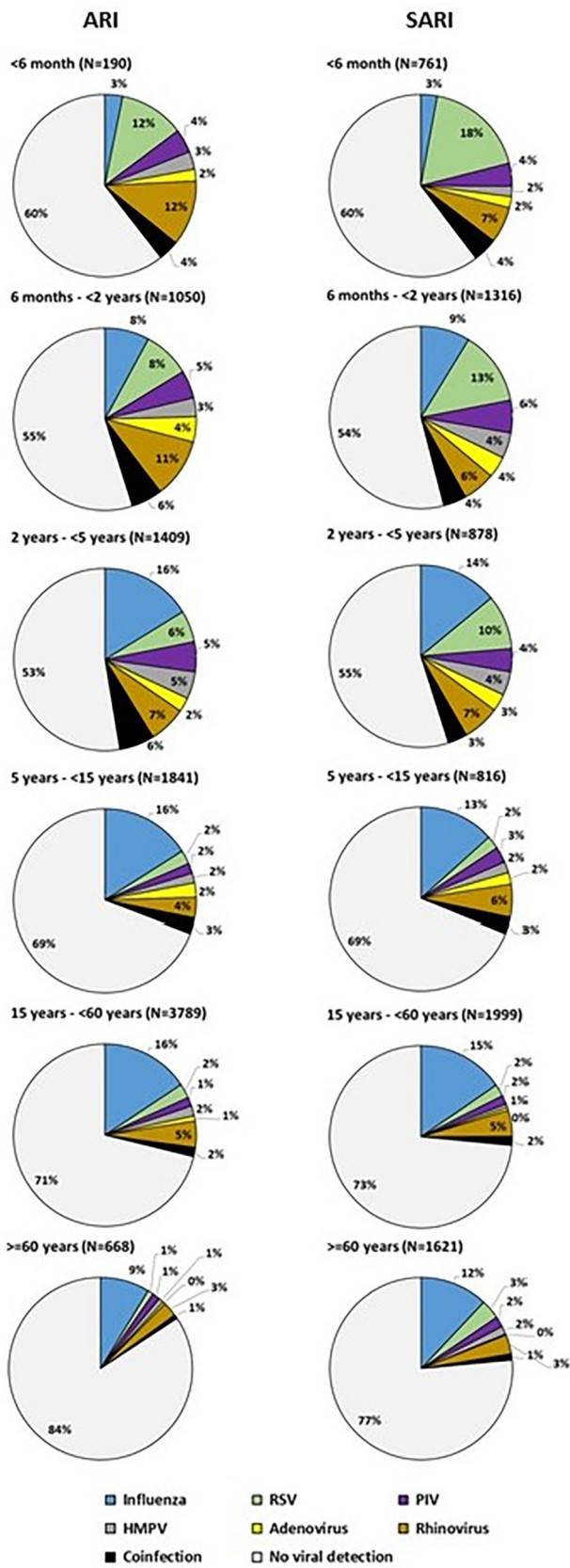

**Fig 1. Distribution of influenza and other respiratory viruses among ARI and SARI cases by age-group in India (2016–18).**

Co-detection of viruses was seen in 269 (3%) of ARI and 190 (2.6%) of SARI cases. HRV were the most frequent virus with co-detection (153/459, 32.6%) followed by AdV (147/459, 31.3%), HMPV (127/459, 27.7%) and RSV-B (121/459, 26.4%). Influenza-RSV co-detection was seen in 74 cases of which 25 were in SARI cases.

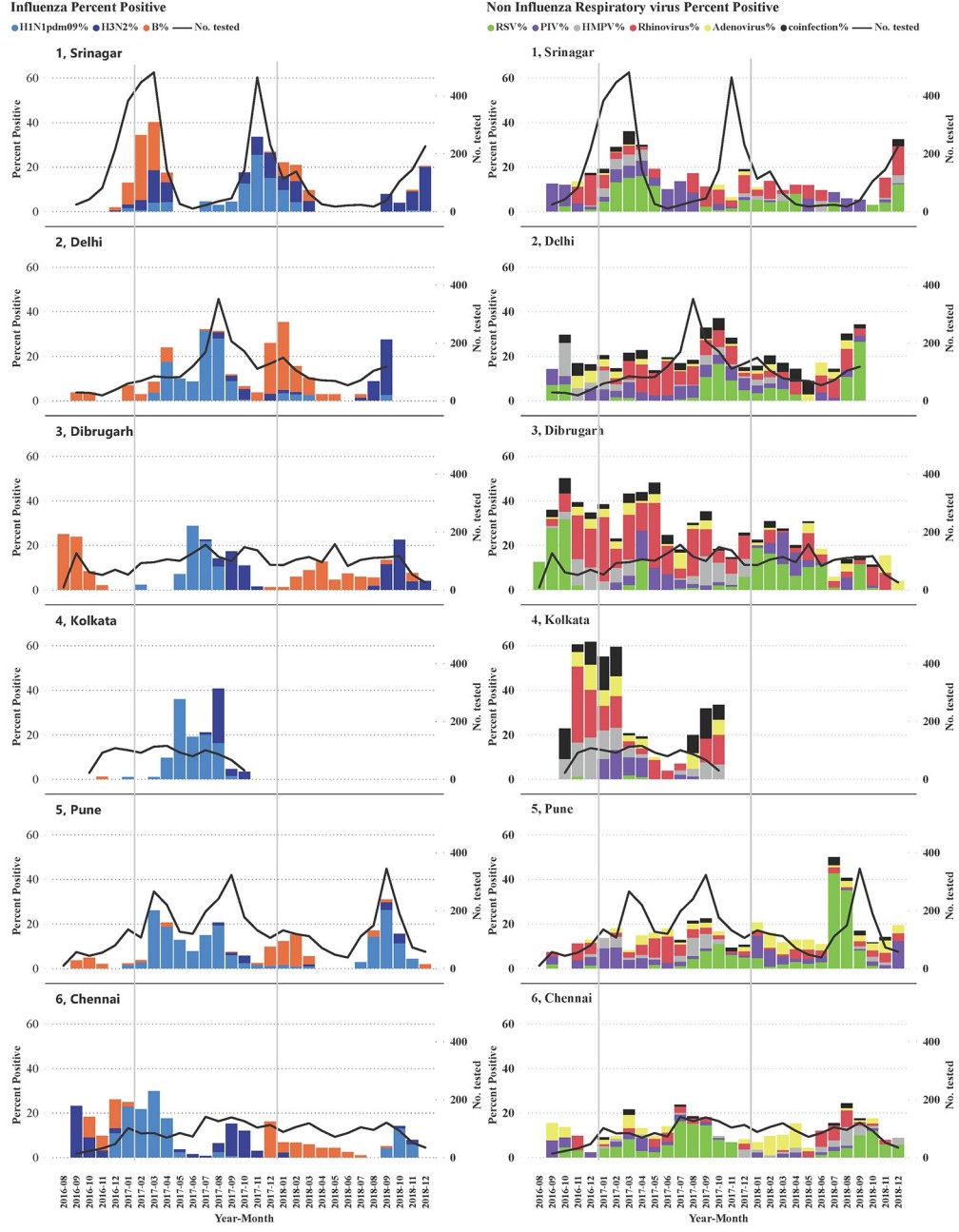

#Legend-In First Column, Light blue in Influenza A(H1N1)pdm, dark blue is influenza A(H3N2) and red is Influenza B. In second column, green is RSV, yellow is adenovirus, purple is PIV, grey is HMPV, red is rhinovirus, and black is coinfection.
Monthly % positivity is the proportion of virus positive out of all ARI and SARI cases with date of onset of symptom in that month

**Fig 2. Monthly percent positivity of influenza and other respiratory viruses at different sites in India (2016–18).**

## Discussion

The results of this multisite surveillance suggest that influenza surveillance capacity can be leveraged for the surveillance of other respiratory viruses. Using the ARI and SARI case definitions among all age-group, we detected one or more respiratory virus in one-third of the specimens using RT-PCR. Influenza viruses were most detected followed by RSV among both ARI and SARI cases.

The overall virus detection (ARI: 33%,SARI: 33%) in our study was lower than studies in other tropical and sub-tropical countries like Vietnam (SARI: 42%), Yemen (SARI: 41%), Thailand (ILI: 45%, SARI: 38%), Bolivia (ILI 50%) and Suriname (ILI 61%; SARI 41%) [14–18]. The viral detection in <15 years (ARI: 40%, SARI 41%) in our study were comparable to detection in the same age group in the Thailand study (ARI: 49%, SARI:41%) [16]. Influenza was the most common virus detected in those aged ≥5 years in both ARI and SARI cases. This was similar to finding from studies in Thailand and Vietnam [14, 16]. High proportion of participants aged ≥5 years (70% in ARI and 60% in SARI) would have contributed to higher prevalence of influenza seen in the current analysis. The variation in viral detection across centers in our study could be due to difference in proportion of cases below 5 and above 60 years in different centers. Centers with lower proportion of under 5 years was (Chennai and Pune) or had higher proportion of adults> = 60 years (Srinagar) had the lower detection of viruses in ARI and SARI cases.

Among SARI cases, influenza virus was detected in 27% of pregnant women, 12.4% of cases with pre-existing chronic diseases and 12.7% adults ≥60 years. Studies from India during the same period had shown high incidence of influenza in pregnant women (68 to 90 per 10,000 pregnant women months) and older adults (influenza associated lower respiratory infection incidence:7.9 per 1000 person years) [19, 20]. High burden of influenza among these WHO Strategic Advisory Group of Experts (SAGE) target groups suggests that value proposition of influenza vaccination and early interventions (early detection and antiviral treatment) among these groups need to be explored by the country.

In contrast to influenza viruses, other respiratory viral pathogens predominated mainly among children under 5 years, consistent with earlier published data on non-influenza viruses [17, 21]. RSV was most commonly detected virus among children less than 2 years of age both among ARI and SARI cases, emphasizing the leading role of RSV in respiratory infections in young children. Globally, it has been shown that lower middle-income countries contribute to nearly 40% RSV associated ALRI cases and nearly 75% of the RSV-related deaths [22, 23].

The increased activity of influenza in Delhi, Kolkata, Dibrugarh, Pune and Chennai corresponded with rainy season (June to September in Delhi, Kolkata, Dibrugarh, Pune; October to December in Chennai) and in Srinagar with winter (December to February). These results were similar to results from our previous study, displaying discrete periods of peak activity [6, 7]. We also observed an inter-seasonal increase in influenza A (H1N1) pdm09 virus detection across all centers between January and July 2017. Other countries in the region, such as Thailand, Sri Lanka and other South-east Asian countries also displayed a relatively heightened influenza A(H1N1)pdm09 activity throughout 2017 [24].

In contrast to developed countries with temperate climates, influenza epidemiological studies are limited in LMIC. Furthermore, data on non-influenza respiratory infections are also limited. Most of the studies in LMIC are focused on specific age groups (children or elderly population) or ILI/SARI cases. Hence, we tried to address the epidemiological and clinical characteristics of influenza and ORVs across all age groups in both ARI and SARI cases in this paper. We demonstrated that the ARI and modified WHO SARI case definition were useful for detecting influenza and other respiratory viruses. India like many countries leveraged the

influenza sentinel surveillance system for COVID-19 pandemic which further underscored the importance for such robust surveillance systems with capacity to detect other respiratory viruses early as well [25]. The survey had several limitations. We have used a convenient sampling method for recruitment, and this could potentially contribute to overestimation of percent positivity due to increased sampling during peak period. Also, considering the vast size, diverse terrain and climates, and large rural populations of India, the data from the six cities may not be generalizable to whole country. Additional years of surveillance data from different geographical regions within the country will contribute to better understanding of epidemiology of ORVs. Although a substantial proportion of ARI and SARI is associated with bacterial pathogens, only respiratory viruses were tested for. Nevertheless, our study was aimed to leverage the existing influenza surveillance sites collecting upper respiratory specimens for ORV surveillance, and therefore was inappropriate for bacterial pathogens.

We demonstrated the usefulness of influenza sentinel surveillance platform to collect comprehensive information on the viral etiology of SARI and ARI cases in a pre-COVID-19 period. Our findings suggest influenza sentinel surveillance systems may be leveraged for surveillance of RSV, SARS-CoV2 and other respiratory viruses.

## Supporting information

**S1 Table. Monthly virus percent positivity by site, India (September 2016 to December 2018).**
(DOCX)

**S1 Data. Data for figures.**
(XLSX)

## Acknowledgments

The authors would like to acknowledge the clinical collaborator of all the sites for identification of cases, sharing clinical data and collecting respiratory samples from the study participants. Authors are grateful to all the project technical staff involved in sampling, testing and reporting. Authors would also like to acknowledge the contribution of Dr DT Mourya (Ex Director, National Institute of Virology, Pune and Dr Seema Jain (Ex Director Influenza program, US Centers for Disease Control and Prevention, India office).

## Author Contributions

**Conceptualization:** Mandeep Chadha, Manohar Lal Choudhary, Dipankar Biswas, Parvaiz Koul, K. Kaveri, Lalit Dar, Chawla Sarkar Mamta, Kayla Laserson, Siddhartha Saha, Varsha Potdar.

**Data curation:** Aslesh Ottapura Prabhakaran, Manohar Lal Choudhary, Dipankar Biswas, Parvaiz Koul, K. Kaveri, Lalit Dar, Chawla Sarkar Mamta, Sumit Dutt Bhardwaj, Siddhartha Saha.

**Formal analysis:** Mandeep Chadha, Aslesh Ottapura Prabhakaran, Manohar Lal Choudhary, Santosh Jadhav, Sumit Dutt Bhardwaj, Siddhartha Saha, Varsha Potdar.

**Funding acquisition:** Mandeep Chadha.

**Investigation:** Mandeep Chadha, Manohar Lal Choudhary, Dipankar Biswas, Parvaiz Koul, K. Kaveri, Lalit Dar, Chawla Sarkar Mamta.

**Methodology:** Mandeep Chadha, Manohar Lal Choudhary, Dipankar Biswas, Parvaiz Koul, K. Kaveri, Lalit Dar, Chawla Sarkar Mamta, Kayla Laserson, Varsha Potdar.

**Project administration:** Mandeep Chadha, Manohar Lal Choudhary, Dipankar Biswas, Parvaiz Koul, K. Kaveri, Lalit Dar, Chawla Sarkar Mamta, Sumit Dutt Bhardwaj, Kayla Laserson, Siddhartha Saha.

**Resources:** Mandeep Chadha, Manohar Lal Choudhary.

**Software:** Manohar Lal Choudhary, Santosh Jadhav, Siddhartha Saha.

**Supervision:** Mandeep Chadha, Aslesh Ottapura Prabhakaran, Manohar Lal Choudhary, Dipankar Biswas, Parvaiz Koul, Chawla Sarkar Mamta, Sumit Dutt Bhardwaj, Kayla Laserson, Siddhartha Saha, Varsha Potdar.

**Validation:** Aslesh Ottapura Prabhakaran, Manohar Lal Choudhary, Santosh Jadhav, Siddhartha Saha, Varsha Potdar.

**Visualization:** Aslesh Ottapura Prabhakaran, Siddhartha Saha.

**Writing – original draft:** Mandeep Chadha, Aslesh Ottapura Prabhakaran, Manohar Lal Choudhary, Sumit Dutt Bhardwaj, Siddhartha Saha, Varsha Potdar.

**Writing – review & editing:** Mandeep Chadha, Aslesh Ottapura Prabhakaran, Manohar Lal Choudhary, Dipankar Biswas, Parvaiz Koul, K. Kaveri, Lalit Dar, Chawla Sarkar Mamta, Santosh Jadhav, Sumit Dutt Bhardwaj, Kayla Laserson, Siddhartha Saha, Varsha Potdar.

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
