## [Decision Letter · Decision Letter 0]

15 Jul 2022

PGPH-D-22-00929

Multisite surveillance for influenza and other respiratory viruses in India: 2016-2018

Dear Dr. Potdar,

Thank you for submitting your manuscript to PLOS Global Public Health. After careful consideration, we feel that it has merit but does not fully meet PLOS Global Public Health’s publication criteria as it currently stands. Therefore, we invite you to submit a revised version of the manuscript that addresses the points raised during the review process.

We look forward to receiving your revised manuscript.

Kind regards,

Raquel Muñiz-Salazar, Ph.D.

Academic Editor

Journal Requirements:

1. Please provide a/amend your detailed Financial Disclosure statement. This is published with the article. It must therefore be completed in full sentences and contain the exact wording you wish to be published.

2. In the online submission form, you indicated that "The corresponding author would provide minimal dataset consisting of variables used in analysis upon request after receiving clearance from Ministry of Health and Family Welfare, India". All PLOS journals now require all data underlying the findings described in their manuscript to be freely available to other researchers, either 1. In a public repository, 2. Within the manuscript itself, or 3. Uploaded as supplementary information.

3. We do not publish any copyright or trademark symbols that usually accompany proprietary names, eg (R), (C), or TM  (e.g. next to drug or reagent names). Please remove all instances of trademark/copyright symbols throughout the text, including SuperScriptTM and Epi-info 7(TM).

4. Please provide separate figure files in .tif or .eps format and removed from the manuscript file.

5. We notice that your supplementary [figures/tables] are included in the manuscript file. Please remove them and upload them with the file type 'Supporting Information'. Please ensure that each Supporting Information file has a legend listed in the manuscript after the references list.

Additional Editor Comments (if provided):

The paper is well-written, the sample size is large (over 16,000), and it covers different geographical regions in the country.

There are only a few comments that must be attended to.

To detail the basic strategies used to identify these cases at the sites and some idea of average daily recruitment.

To discuss site-specific differences and limitations on the use of ARI/SARI case definition for different viruses.

In Tables 1 & 2, for the OR estimation for different symptoms, it is not clear which is the reference category.

The font appears inconsistent between the main text and references. Also, spacing in the funding and acknowledgment section appears single and different from the sections above.

Reviewers' comments:

Reviewer's Responses to Questions

**Comments to the Author**

1. Does this manuscript meet PLOS Global Public Health’s publication criteria? Is the manuscript technically sound, and do the data support the conclusions? The manuscript must describe methodologically and ethically rigorous research with conclusions that are appropriately drawn based on the data presented.

Reviewer #1: Yes

Reviewer #2: Yes

2. Has the statistical analysis been performed appropriately and rigorously?

Reviewer #1: Yes

Reviewer #2: Yes

3. Have the authors made all data underlying the findings in their manuscript fully available (please refer to the Data Availability Statement at the start of the manuscript PDF file)?

Reviewer #1: Yes

Reviewer #2: No

4. Is the manuscript presented in an intelligible fashion and written in standard English?

Reviewer #1: Yes

Reviewer #2: Yes

5. Review Comments to the Author

Reviewer #1: The paper is well-written and shows team effort from experienced researchers. It is a prospective project involving 6 selected sites based on climate and geography. the sample size of over 16 000 could be better for such a populous country. Nevertheless, it still represents the 6 sites well enough for a quantitative study. Results are from a primary research which is scientifically sound and analyzed properly. This is a pre-COVID-19 project, and the picture could have changed drastically since then. The project period ran between 5 years to 3 years ago. However, the message is consistent with the conclusion and is still relevant. Data availability is dependent upon clearance to use dataset of variables used in the data analysis by the ministry of health and family welfare in India. The corresponding author would avail such information. There is no obvious evidence that the work is similar in grammar and content to any which has been published before. In general the font appears inconsistent between the main text and references. Also, spacing in the funding and acknowledgement section appears single and different from sections above. This must be attended to.

Reviewer #2: The paper presents positivity rates by different respiratory viruses for about two years for ARI and SARI and is a useful addition especially from the point of view of other resp viruses. The sample size is large and covers different geographical regions in the country.

One key concern would be the convenient sample of the cases within the selected facilities (which are also convenient but probably non-modifiable). This is pointed out in the limitation. However, it would be good to detail the actual strategies used to identify these cases at the sites (first five?) as well as some idea of average daily recruitment.

Suggest use the term percent positivity rather than period prevalence throughout the paper. The use of High mPP based on average is not really useful and supplementary table is difficult to read also as months gets repeated horizontally.

Some more discussion on site-specific differences as well as limitations on use of ARI/SARI case definition for different viruses will add more insight to the paper.

In table 1 & 2, for the OR estimation for different symptoms, it is not clear which is the reference category.

6. PLOS authors have the option to publish the peer review history of their article (what does this mean?). If published, this will include your full peer review and any attached files.

**Do you want your identity to be public for this peer review?** For information about this choice, including consent withdrawal, please see our Privacy Policy.

Reviewer #1: **Yes: **John Mukuka Musonda

Reviewer #2: **Yes: **Anand Krishnan

---

## [Decision Letter · Decision Letter 1]

5 Oct 2022

Multisite surveillance for influenza and other respiratory viruses in India: 2016-2018

PGPH-D-22-00929R1

Dear Dr Potdar,

We are pleased to inform you that your manuscript 'Multisite surveillance for influenza and other respiratory viruses in India: 2016-2018' has been provisionally accepted for publication in PLOS Global Public Health.

Best regards,

Raquel Muñiz-Salazar, Ph.D.

Academic Editor

Taking the reviewers' statement and because all comments have been addressed. I recommend ACCEPT the manuscript.

Reviewer Comments (if any, and for reference):

Reviewer's Responses to Questions

**Comments to the Author**

1. If the authors have adequately addressed your comments raised in a previous round of review and you feel that this manuscript is now acceptable for publication, you may indicate that here to bypass the “Comments to the Author” section, enter your conflict of interest statement in the “Confidential to Editor” section, and submit your "Accept" recommendation.

Reviewer #1: All comments have been addressed

2. Does this manuscript meet PLOS Global Public Health’s publication criteria? Is the manuscript technically sound, and do the data support the conclusions? The manuscript must describe methodologically and ethically rigorous research with conclusions that are appropriately drawn based on the data presented.

Reviewer #1: Yes

3. Has the statistical analysis been performed appropriately and rigorously?

Reviewer #1: Yes

4. Have the authors made all data underlying the findings in their manuscript fully available (please refer to the Data Availability Statement at the start of the manuscript PDF file)?

Reviewer #1: Yes

5. Is the manuscript presented in an intelligible fashion and written in standard English?

Reviewer #1: Yes

6. Review Comments to the Author

Reviewer #1: Concerns from the previous round have been addressed adequately.

7. PLOS authors have the option to publish the peer review history of their article (what does this mean?). If published, this will include your full peer review and any attached files.

**Do you want your identity to be public for this peer review?** For information about this choice, including consent withdrawal, please see our Privacy Policy.

Reviewer #1: **Yes: **John Mukuka Musonda
